# Deep Edge Detection Methods for the Automatic Calculation of the Breast Contour

**DOI:** 10.3390/bioengineering10040401

**Published:** 2023-03-24

**Authors:** Nuno Freitas, Daniel Silva, Carlos Mavioso, Maria J. Cardoso, Jaime S. Cardoso

**Affiliations:** 1Faculdade de Engenharia, Universidade do Porto, 4099-002 Porto, Portugal; 2INESC TEC, 4200-465 Porto, Portugal; 3Breast Unit, Champalimaud Foundation, 1400-038 Lisbon, Portugal; 4Faculty of Medicine, University of Lisbon, 1649-004 Lisbon, Portugal

**Keywords:** breast cancer, aesthetic assessment of breast cancer surgery outcomes, artificial intelligence, breast cancer conservative treatment, edge detection, computer vision

## Abstract

Breast cancer conservative treatment (BCCT) is a form of treatment commonly used for patients with early breast cancer. This procedure consists of removing the cancer and a small margin of surrounding tissue, while leaving the healthy tissue intact. In recent years, this procedure has become increasingly common due to identical survival rates and better cosmetic outcomes than other alternatives. Although significant research has been conducted on BCCT, there is no gold-standard for evaluating the aesthetic results of the treatment. Recent works have proposed the automatic classification of cosmetic results based on breast features extracted from digital photographs. The computation of most of these features requires the representation of the breast contour, which becomes key to the aesthetic evaluation of BCCT. State-of-the-art methods use conventional image processing tools that automatically detect breast contours based on the shortest path applied to the Sobel filter result in a 2D digital photograph of the patient. However, because the Sobel filter is a general edge detector, it treats edges indistinguishably, i.e., it detects too many edges that are not relevant to breast contour detection and too few weak breast contours. In this paper, we propose an improvement to this method that replaces the Sobel filter with a novel neural network solution to improve breast contour detection based on the shortest path. The proposed solution learns effective representations for the edges between the breasts and the torso wall. We obtain state of the art results on a dataset that was used for developing previous models. Furthermore, we tested these models on a new dataset that contains more variable photographs and show that this new approach shows better generalization capabilities as the previously developed deep models do not perform so well when faced with a different dataset for testing. The main contribution of this paper is to further improve the capabilities of models that perform the objective classification of BCCT aesthetic results automatically by improving upon the current standard technique for detecting breast contours in digital photographs. To that end, the models introduced are simple to train and test on new datasets which makes this approach easily reproducible.

## 1. Introduction

Breast cancer is the most frequently diagnosed form of cancer in women worldwide. According to recent data [1], the incidence of this type of cancer has been rising. Despite that, the mortality rate has been steadily decreasing in the past years, leading to an increased interest in the patient’s Quality of Life (QoL) after treatment [2].

Breast cancer conservative treatment (BCCT) has become a frequent alternative to mastectomy as it achieves similar survival rates while improving the cosmetic outcome. This procedure consists of removing the tumor and a small area of surrounding tissue and treating the remaining tissue with radiotherapy.

Currently, there is still no gold-standard for evaluating the cosmetic results of the operation objectively. The aesthetic classification of breast cancer surgery is usually done according to the Harvard scale introduced by Jay Harris in 1979 [3]. This method separates BCCT aesthetic outcomes into 4 categories: excellent, good, fair, and poor. To classify a subject into these classes there are several valid approaches, both subjective and objective. The most common way of evaluating the results is the subjective appreciation of the patient. However, considering the subjectivity of the human evaluation, this has not led to satisfying results [2]. An initial approach to combat this problem was to compare measurements between the two breasts, either directly in patients or in digital photographs.

Fitzal et al. [4] and Cardoso et al. [5] proposed Computer-Aided Aesthetic objective classification methods for evaluating the cosmetic outcome of BCCT based on breast features identifiable on digital images of the patient. None were fully automatic and both needed some manual annotation for the extraction of features.

Fitzal developed the Breast Analysing Tool that measures the Breast Symmetry Index (BSI) to assess the cosmetic outcome. Cardoso developed BCCT core, a tool that performed automatic feature extraction from digital photographs to classify cosmetic outcomes of BCCT. However, to use this tool, it was still necessary to manually mark some keypoints in the image such as the left and right nipple and breast contours. Hence, it is relevant to have good automatic methods for the extraction of breast features.

Despite advances in recent years, current models are still inadequate for automatic detection of breast contours, especially when applied to photographs taken in unseen environments during model development. Changes in brightness, colour saturation, perspective, or background in a photograph should not affect the prediction of these models. An ideal algorithm should work robustly with data from different breast cancer units and perform flawless detection of breast contours to automate the aesthetic evaluation of BCCT.

The main contribution of this work is a new method for breast contour detection that is robust enough to deal with different scenarios that differ from its training standpoint. The idea was to develop a model that implicitly derives its breast contour detection task and performs it as intended, regardless of changes in perspective or background in the images. This was achieved by starting from the concept that breast contour detection is a special kind of edge detection.The methods developed convert the traditional Shortest Path method into a deep learning approach that can learn intuitively from data and can improve upon this method while maintaining its generalization capabilities.

In addition to the introduction, this paper is organized as follows: Section 2 analyzes the state-of-the-art method developed on the same topic, and Section 3 explains our proposed method. Section 4 shows the experimental implementation of this method and the obtained results. The results are discussed in Section 5, and Section 6 presents a more comprehensive study including different edge operators for the shortest-path method. Finally, the conclusion in Section 7 contains an appreciation of the obtained results and the overall contribution of this work.

## 2. Previous Work

In 2007, Cardoso and Cardoso developed a computer-aided medical system capable of automatically providing an objective measure of the aesthetic result of BCCT [5]. This system relied on the extraction of various features from digital photographs of the patient’s torso. In this system, some features had to be manually annotated from the photographs. Further work on this topic tapped the automatic extraction of relevant features using conventional, full deep learning and hybrid approaches. The following works focus on the development of automatic methods for the detection of breast contours.

### 2.1. Conventional Method: A Shortest-Path Approach to Contour Detection

In 2008, Cardoso et al. proposed an automatic method for breast contour detection in BCCT patients based on 2D digital torso images [6]. Given breast contour endpoints, breast contour search can be abstracted as the problem of finding the shortest path connecting them as long as the breast interior is free of edges.

For this purpose, it is necessary to model the image as a graph-like object, where pixels representing edges in the original image are preferred in path selection. In this method, a conventional Sobel operator was used to detect edges converted into a weighted graph where each node represents a pixel and the weight of each arc can be calculated based on the pixels connecting them (an arc connecting two strong edge pixels has a lower weight, while an arc connecting two weak non-edge pixels receives a higher weight).
(1)f(g)=α×exp(β(255−g))+γ,
(2)$$w_{i,j}=\left\{\begin{array}{c} f(g),\;\mathrm{if}\;i\;\mathrm{and}\;j\;\mathrm{are}\;\mathrm{vertically}\;\mathrm{or}\;\mathrm{horizontally}\;\mathrm{adjacent}\\ \sqrt{2}f\left(g\right),\;\mathrm{if}\;i\;\mathrm{and}\;j\;\mathrm{are}\;\mathrm{diagonally}\;\mathrm{adjacent} \end{array}\right.\;.$$

Equations (Equation 1) and (Equation 2) show the mathematical expression for calculating arc weights. α, β, and γ are adjustable parameters, while g is the average of the Sobel gradient on two incident pixels. $w_{i,j}$ represents the arc weight connecting the adjacent nodes *i* and *j*. Diagonally adjacent nodes are considered to be further apart, and so, their weight is multiplied by $\sqrt{2}$.

Finally, Dijkstra’s algorithm [7] is used to find the shortest path between the two endpoints in this weighted graph. The cost of a path was defined as the sum of all arcs connecting its nodes (Equation (Equation 3)).
(3)$$Cost_{path}=\sum w_{i,j},\;\mathrm{where}\;\left(i,j\right)\;\mathrm{are}\;\mathrm{pairs}\;\mathrm{of}\;\mathrm{adjacent}\;\mathrm{nodes}\;\mathrm{in}\;\mathrm{the}\;\mathrm{path}.$$

This method was designed and tested on a small, homogenous dataset. Nevertheless, it achieved positive results.

This work was further improved by Sousa et al. [8] with prior shape modeling. The typical round or teardrop shape of a breast meant that given any two breast endpoints, some areas of the image had a higher probability of belonging to the breast contour. This probability mask is applied to the weighted graph to enforce the higher likelihood paths, showing positive results.

### 2.2. Deep Methods

Silva et al. proposed a method that uses a Deep Neural Network (DNN) for automatic keypoint detection in patients’ torso photographs [9]. This DNN holds two modules: Heatmap Regression and Refinement and Keypoint Regression. The first module uses the U-Net model to learn how to regress the ground truth keypoints from heatmaps obtained from Gaussian filtering. These heatmaps follow a recurrent process, with each input corresponding to the multiplication between the original image and the output from the previous heatmap. The keypoint regression takes the output from the latter module and applies the VGG16 model [10] to obtain the final coordinates.

Gonçalves et al. combined deep image segmentation [11] and the model proposed by Silva et al. The U-Net ++ architecture [12] was trained to generate breast segmentation masks. These masks were then used to extract contours using the marching squares algorithm. As a post-processing step, the predicted endpoints by Silva et al. are projected onto the mask contours to generate the desired breast contour keypoints.

### 2.3. Hybrid Method

Silva et al. also proposed a hybrid approach to keypoint detection in their original work, combining deep learning methods with the traditional shortest-path algorithm [9]. In this approach, the deep keypoint detection algorithm provides the endpoints that are used as input to the shortest-path algorithm. This method improved the results obtained of the full deep model.

### 2.4. Deep Edge Detection: RINDNet

In 2021, Pu et al. presented a new model [13] of image edge detection capable of detecting four types of edges simultaneously. RINDNet is an end-to-end model that works in three stages (see Figure 1).

Stage I captures general features and spatial cues for all edges. Stage II divides the model into four separate decoders while separating low-level and high-level features. Stage III predicts the separate results for each edge type. The four different edge types proposed are Reflectance, Illumination, Normal and Depth. This approach is useful for our application as it can be trained to emphasize breast contour while removing unwanted noise from other edges.

A recent survey by Sun et al. [14] has classified RINDNet as one of the state-of-the-art models for edge detection, as it showed competitive performance on two different datasets. Furthermore, this model’s ability to differentiate between edge types makes it an interesting choice for our application, as we want to focus the edge detection on the breast contour edges. In our proposal, a final 1×1 convolutional layer combines these four predicted edge types into a single-edge gradient map.

## 3. Proposed Method

The method proposed in this paper follows a hybrid approach of conventional and deep methods, with two modules: Deep Edge Detection and shortest-path Estimation. This approach follows a rationale similar to the conventional method developed by Cardoso et al. [6], but replaces the Sobel Edge operator with deep learning models that can be optimized for the task at hand. For the Deep Edge Detection step, the main goal was to obtain a simple model that provides edges between skin regions and integrates more semantic information into the detection phase than is possible with Sobel-like operators, while keeping the shortest-path algorithm the same as that proposed by Cardoso et al. [6].

Figure 2 shows a diagram of the proposed model where the algorithm is broken down into three steps:Edge Detection: Use either RindNet or Residual Unet to detect edges in the image;Convert to weighted graph: convert the output of the edge detection step to a graph-like object following Equation 1 and apply the shape priors proposed by Sousa et al. [8];Shortest-path Estimation: Use Dijkstra’s algorithm to compute the shortest path between both endpoints on the weighted graph.

Since the Deep Edge Detection model’s purpose is to produce a weighted graph to be used as the input for the shortest-path algorithm, its output should emphasize breast edges.

The following subsections present the details about the proposed methodology. It is important to note that our proposal requires the Breast Endpoints annotations as inputs, while the methods proposed by Gonçalves et al. [11] and Silva et al. [9] do not. However, we show that our method is more robust to different data sources, while the fully automated methods are too sensitive to perform well across different datasets.

### 3.1. Deep Edge Detection

We investigate two alternatives for the Deep Edge detection phase (Figure 2 step 1): RINDNet (see Section 2) and Sobel U-NET.

#### 3.1.1. SobelU-Net

As discussed in Section 2, the early work revolved around the use of edge operators, such as Sobel filters, for edge detection. These conventional methods can be applied to any image, perform as intended, and require no training. This is an advantage over deep learning methods, where simple variations in the data distribution severely affect the expected results.

Our second alternative for the edge detection block, SobelU-Net, incorporates Sobel’s filters into its architecture. The pipeline follows the basic U-Net model and includes an accompanying encoding path that starts with the Sobel operation result of the original image, as shown in Figure 3.

The original image is encoded using four sets of convolutional blocks, composed of two 3×3 convolutional operations followed by max pooling. After each convolutional block, the number of features increases until it reaches the bottleneck stage, with 256 features. The decoding pathway takes the bottleneck representations and performs four iterations of convolutional blocks followed by transposed convolution. In each iteration, it concatenates and crops the decoded rendering with the encoded Sobel operation.

#### 3.1.2. Model Comparison

These models were selected to determine the appropriate level of complexity for this application. RINDNet is a state-of-the-art, complex, and computationally intensive edge model (with over 59 million parameters and a size of 5.5 × 10${}^{13}$ MegaBytes) that is suitable for complex problems and can learn and refine specific edge detection. To investigate whether the complexity of RINDNet is necessary for our specific task, we selected a second model with contrasting properties. SobelU-NET is a simple and lightweight model (over 1.6 million parameters and 911 MegaBytes in size) chosen to understand whether a model as advanced as RINDNet is necessary for this application.

Figure 4 shows examples of the output of the trained models and the Sobel Filter. It is visible that these models present the capabilities of focusing more on specific types of edges that are important to our application. Figure 5 shows the predicted path for the same image using all three options (Sobel U-NET, RINDNet and Sobel); this output further shows how the models are capable of strengthening breast skin edges to improve results.

### 3.2. Model Optimization

The datasets provided consisted of digital photographs of the patients’ torsos and annotations of only the breast contours (these annotations were obtained by having the annotator define a series of keypoints in the breast contour and then determining the shortest path between the endpoints passing through the keypoints, see Figure 6). It was necessary to develop a training process capable of refining the edge detection models for the breast contour using the available weak annotation.

Let *I* be an image in the dataset and gt(I) be the annotated groundtruth path, a sequence of connected pixels ($x\in R^{2}$ connected positions) between a starting pixel $x_{s}$ and an ending pixel $x_{e}$.

In addition, let w be the set of all parameters of our neural network and f(I;w) be the (edge) map outputted by the NN.

The cost of the groundtruth path in the current edge map can be represented as $c_{gt}\left(f\left(I;w\right),gt\left(I\right)\right)$. For the current edge map, we also compute the predicted shortest path between endpoints $x_{s}$ and $x_{e}$, pp(f(I;w); $x_{s}$, $x_{e}$) as a sequence of connected pixels ($x\in R^{2}$ connected positions).

The goal is to get the NN to learn an edge representation where the groundtruth path is the same as the shortest path (and where the cost is also the same). Our loss function (Equation (Equation 4)) accounts for this by penalizing NNs where the cost of the true path is higher than the cost of the shortest path.
(4)$$L\left(I;w,x_{s},x_{e}\right)=ReLU\left(cost(gt;w)-cost(pp;w)\right)$$

It is easy to understand that this loss tends to zero when the cost of the ground truth path is similar to the predicted path cost. The goal of the training process is to minimize this loss, which should result in increased similarity between the predicted and the ground truth paths.

## 4. Experimental Evaluation

The proposed models were trained and tested on a mixed dataset of 221 images that joins images from three smaller datasets: the first one (dataset 1) is composed of 120 photographs of patients where the torso of the patient is shown against a clean, uniform background. The second (dataset 2) contained 30 other photographs obtained in similar conditions. The third and final dataset (dataset 3) was composed of 71 images with poorer lighting and no consistency in the background.

In addition, we used a new database for testing purposes only (referred to in this document as the BCCT core database) that contained images from a single breast care unit where acquisition conditions were not tightly controlled. This dataset was used for testing only, to assess the ability of the models to fit new datasets. Both databases are public and can be accessed upon request to the authors.

Figure 7 and Figure 8 show some examples of these databases. It can be seen that in the BCCT core database, the centering and distance of the patient’s torso from the camera were much more variable, which presented much greater challenges to the models.

Our proposed models are compared with the deep learning models described in Section 2. It is worth noting that we divide the models into Automatic and Semi-Automatic, depending on whether or not they require endpoint annotation to predict the breast contours (see Figure 6).

For the mix dataset, the models were trained and tested using 5-fold cross-validation. The baseline results were obtained by running the algorithm with the Sobel edge detection across all images. Table 1 shows the average results of the five test folds: mean error, standard deviation, and maximum error in the fold respectively.

It can be seen that both the SobelU-Net model and the RINDNet model outperform the Baseline model. The best results for mean error were obtained with the RINDNet convolutional model, while the Segmentation model had the lowest maximum error per fold and the lowest standard deviation.

Figure 9 shows the images and contour detection results for the worst possible case obtained with one of the trained RindNet models. It can be seen that RindNet outperforms the Sobel results even in the worst case error.

To obtain a more sophisticated scenario, we evaluated performance in a cross-dataset setup. Therefore, we trained the models on the mix dataset and tested them on the BCCT core dataset, which contains more diverse images. Table 2 shows the results for the BCCT core dataset. We find that the fully automatic deep models previously proposed in the literature do not perform well when faced with a different dataset than the one they were trained on. This effect points to model overfitting to the training data.

In this experiment, SobelU-NET outperforms all other models on all measurements. Figure 10 shows one of the cases where the pre-trained SobelU-NET model is able to improve the results of the conventional approach. In most cases, the use of the Shortest Path model with the Sobel edge detector already leads to positive results. In this sense, Sobel U-NET achieves a similar performance, but it improves performance when the Sobel version’s shortest path leads to unsatisfactory results. This can be concluded from Table 2 as the maximum error decreases significantly. RINDNet does not perform as well in this experiment, failing to surpass the Sobel mean error. This is because the model overfits to the training data and does not show equivalent performance in the new dataset. However, it still performs much better than the Automatic Deep methods.

## 5. Discussion

As shown in Table 1 and Table 2, the Automatic Deep methods perform well in the dataset they were trained in, but they present unstable results when faced with testing data from different sources and photographs obtained under variable conditions. In contrast, the Semi-Automatic methods perform well in both experiments, especially SobelU-NET. This model surpasses the Baseline model across all measurements, outperforming RINDNet in the cross-dataset experiment. This can be justified as the UNET is a less complex Deep Model that is more faithful to the original Sobel approach. Hence, it does not overfit the training dataset. Even though it does not perform as well in the original mix dataset, it is a more robust model capable of performing well under variable conditions.

These results indicate the SobelU-NET model is the best performing model, making it a useful model with small datasets, as its simplicity when compared to other Deep Models prevents overfitting. On the other hand, if the RINDNet model was to be trained in a more diverse dataset, it could outperform other models as it does on the mix dataset. In future applications, this model should be considered if there is a more adequate and diverse dataset for training and assurance that new datapoints will not have such variable conditions.

## 6. Ablation Experiments

To secure the validity of these results, we experimented with several different edge operators to show how our model perfects edge detection in our application. As the experiments done with previous deep learning models show that those do not surpass the shortest-path approach in this application, we focused on having different conventional and state-of-the-art edge operators and assessing how they perform on our dataset using the shortest-path method.

To that end, we tested two commonly used edge detectors: the Canny edge detection algorithm and the Prewitt operator. Furthermore, we also tried a linear combination of the four outputs of the state-of-the-art RINDNet model, as well as a two output combination that uses the normal and depth edge detection channels, hence reinforcing the edges that should be more important. All these tests performed a 5-fold cross-validation on the 221 image dataset described in Section 4.

As is visible in Table 3, none of these edge detectors surpass the Baseline Sobel filter, with the Canny filter presenting a similar performance. Looking at Section 4, we see that none of these methods surpass our proposed methods.

## 7. Conclusions

In this paper, we propose a method to improve the automatic detection of breast contours in digital images. By replacing the standard Sobel edge detection with a deep learning model, we achieved better performance, even with a small dataset for training.

The original shortest-path approach achieved good results using conventional edge detection. However, it presented poor performance under specific data points where the breast contour edges were not as pronounced. On the other hand, the fully automatic Deep models presented very positive results in the dataset they were trained on, but failed to perform when faced with a new dataset. In light of that, our approach couples the advantages of conventional edge detection and deep learning. By training Deep Edge detection models used for the shortest path calculation, the resulting models are able to correct difficult examples where the conventional model fails while also reducing the tendency for overfitting, as it is still bounded by the conventional shortest-path approach for breast contour detection.

We show this by testing all models on a different dataset than the one they were trained on. Here, only the shortest-path-based models show positive results. The all-around best-performing model across all experiments is the SobelU-NET model. Despite not achieving the best results in the 5-fold cross-validation in the 221 images dataset, SobelU-NET surpasses all other models when experimenting with cross-dataset validation. Subsequently, the model has better generalization capabilities, concealed by incorporating edge filtering operations in its pipeline that adapt to different datasets. Additionally, this model is easy to implement and re-train, requiring only regular hyperparameter tuning, making this process easy to adapt to new datasets. The main contribution of this paper is an application-oriented solution that is a hybrid approach between conventional methods and deep learning. This approach is capable of achieving state-of-the-art results while still maintaining good generalization capabilities. Given the limited datasets available for training and testing, this hybrid approach is a step forward from other solutions as it takes knowledge from the conventional shortest-path approach.

This work provides an immediate solution to breast contour by outlining a more transparent, robust deep learning method built upon the traditional shortest-path approach. In the future, if working with more comprehensive datasets, it might be worth revisiting the deep learning methods proposed by Gonçalves et al. as well as using Graph Convolution Networks (GCN) to substitute the shortest path estimation [15]. However, at the moment, this solution is the most effective as it achieves good performance overall and is capable of learning effectively from the available data.

## Figures and Tables

**Figure 1 bioengineering-10-00401-f001:**
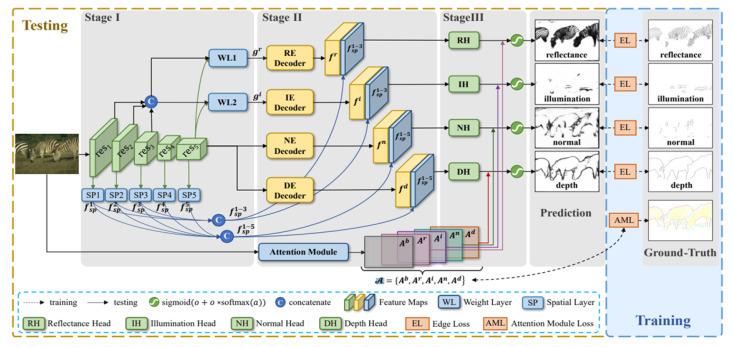
Architecture of RINDNET (from [13]).

**Figure 2 bioengineering-10-00401-f002:**
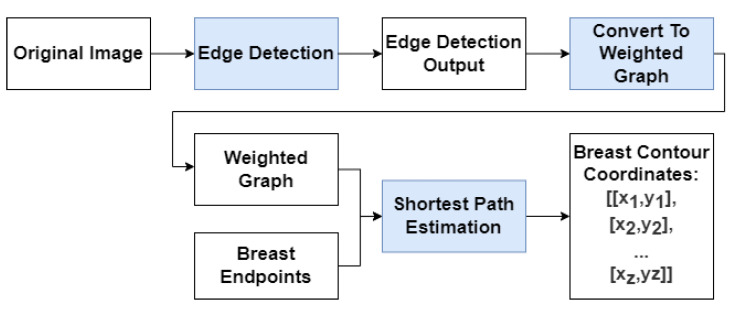
Diagram of the Breast Contour Detection algorithm.

**Figure 3 bioengineering-10-00401-f003:**
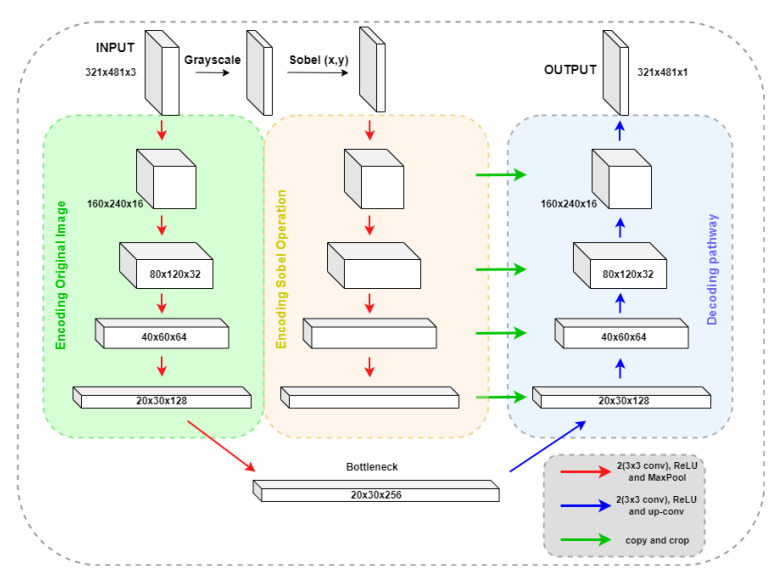
Architecture of SobelU-Net.

**Figure 4 bioengineering-10-00401-f004:**
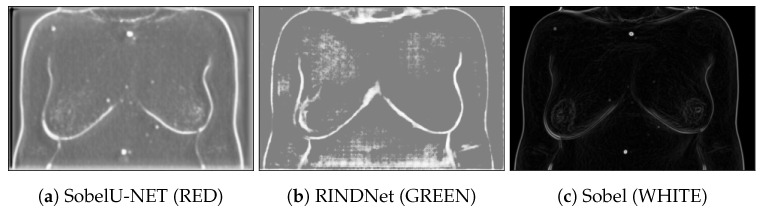
Example of the output of Sobel U-NET (**a**) RINDNet (**b**) and the Sobel Filter (**c**).

**Figure 5 bioengineering-10-00401-f005:**
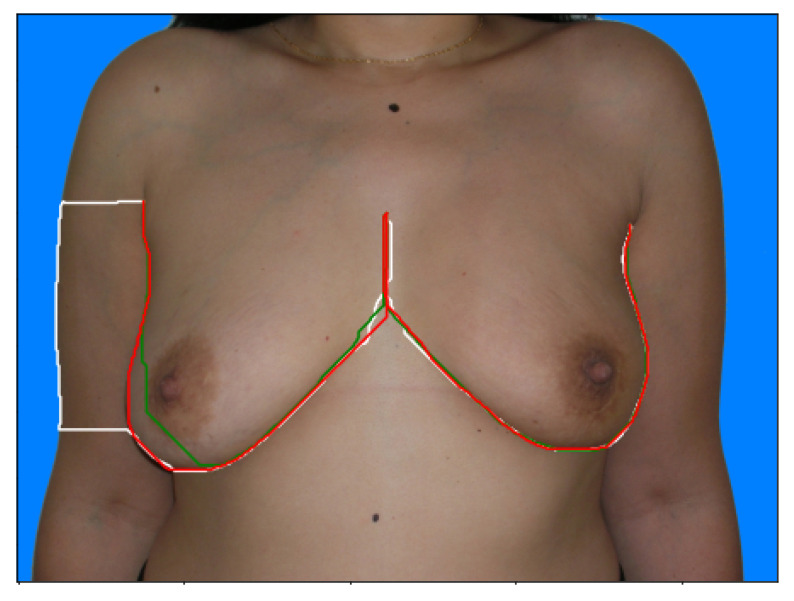
Original Image of Figure 4 with the shortest-path predictions using Sobel U-Net (Red line), RINDNet (Green Line) and the Sobel filter (White line).

**Figure 6 bioengineering-10-00401-f006:**
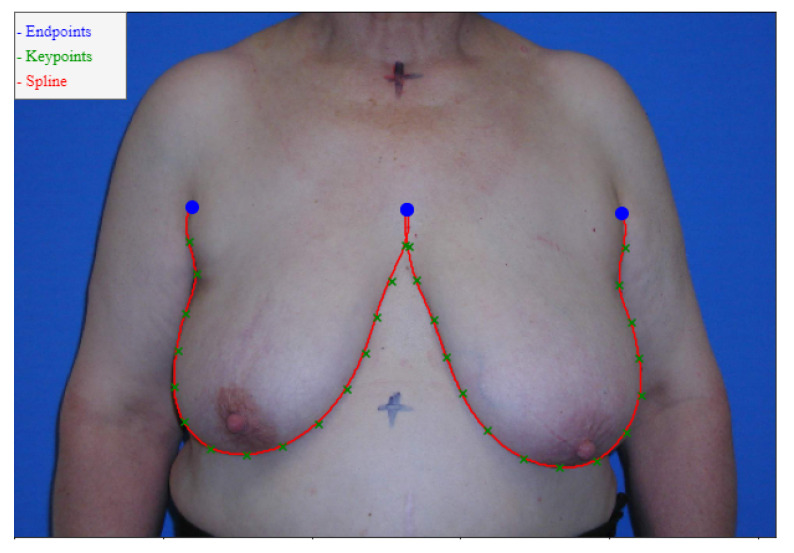
Breast Keypoints (green), Endpoints (blue) and Paths (red) notation example.

**Figure 7 bioengineering-10-00401-f007:**
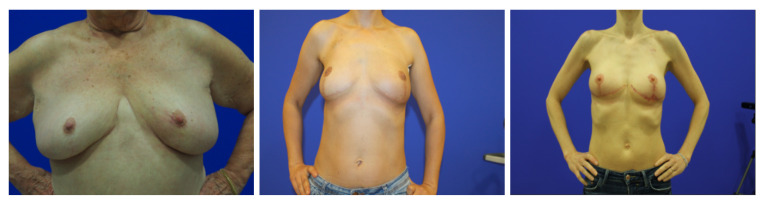
Examples of Images from the BCCT core dataset.

**Figure 8 bioengineering-10-00401-f008:**
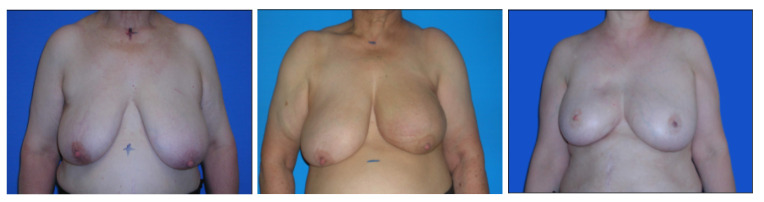
Examples of Images from the mixed dataset of 221 photographs.

**Figure 9 bioengineering-10-00401-f009:**
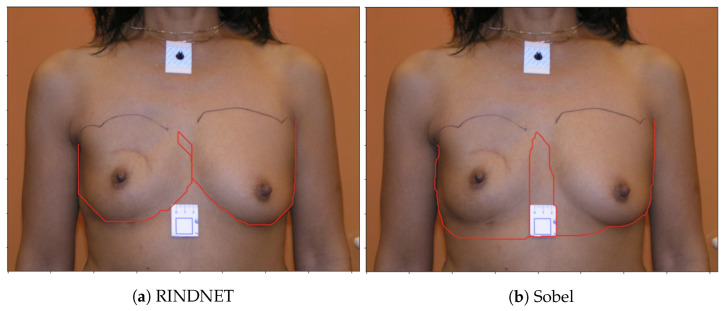
Predicted Breast Contours for shortest-path algorithm using RindNet (**a**) and Sobel (**b**) on the 221 database.

**Figure 10 bioengineering-10-00401-f010:**
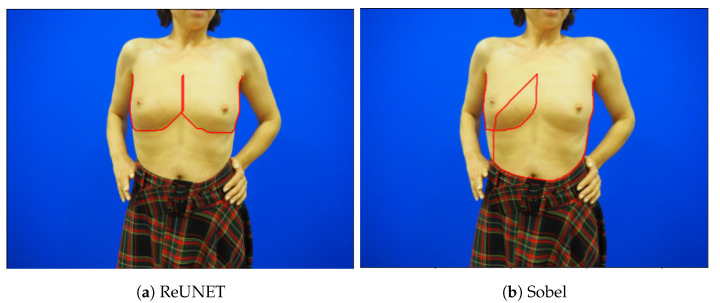
Predicted keypoints for shortest-path algorithm using SobelU-NET (**a**) and Sobel (**b**) on the BCCT core database.

**Table 1 bioengineering-10-00401-t001:** Percentage of the mean, standard deviation and maximum error distance measured in pixels and standardized by the image diagonal. Results are for 5-fold-cross-validation on the mix dataset. Best results are highlighted in bold.

Model	Mean (%)	Std Dev (%)	Max (%)	Type
**Sobel**	0.36	0.71	7.08	Semi-Automatic
**RINDNet**	**0.28**	0.30	2.16	Semi-Automatic
**SobelU-NET**	0.31	0.39	2.83	Semi-Automatic
**DNN Model ***	0.76	0.29	2.61	Automatic
**Seg Model ***	0.40	**0.18**	**1.23**	Automatic

* are results adapted from a paper by Gonçalves et al. [11].

**Table 2 bioengineering-10-00401-t002:** Percentage of the mean, standard deviation and maximum error distance measured in pixels and standardized by the image diagonal. Results are for testing on the BCCT dataset. Best results are highlighted in bold.

Model	Mean (%)	Std Dev (%)	Max (%)	Type
**Sobel**	0.42	0.99	8.80	Semi-Automatic
**RINDNet**	0.48	0.75	8.36	Semi-Automatic
**SobelU-NET**	**0.36**	**0.56**	**4.88**	Semi-Automatic
**DNN Model**	10.97	3.78	23.94	Automatic
**Seg Model**	4.64	3.29	23.56	Automatic

**Table 3 bioengineering-10-00401-t003:** Percentage of the mean, standard deviation and maximum error distance measured in pixels and standardized by the image diagonal. Results are for 5-fold-cross-validation on the mix dataset. Best results are highlighted in bold.

Model	Mean (%)	Std Dev (%)	Max (%)	Type
**Sobel**	**0.38**	0.65	**4.02**	Conventional Model
**Canny**	**0.38**	**0.63**	4.52	Conventional Model
**Prewitt**	0.67	0.88	5.25	Conventional Model
**RINDNet 4 channels**	0.70	0.86	5.53	Deep Model
**RINDNet 2 channels**	0.70	0.97	5.75	Deep Model

## Data Availability

Not applicable.

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
