# Peer review of "Deep Edge Detection Methods for the Automatic Calculation of the Breast Contour"

_bioengineering, 2023, doi:10.3390/bioengineering10040401_

Round 1
Reviewer 1 Report
This paper presents a DL method to improve the capabilities of models that perform the objective classification of BCCT aesthetic results automatically by improving upon the current standard technique for detecting breast contours in digital photographs.
The paper is clearly presented and the methodology has innovation merit especially by employing graph techniques in conjunction with the Sobel coefficient. Some points to be addressed
a) Is the used database publicly available?
b) How can Graph Convoluted Networks (GCN) be employed in this problem in the future?
c) Please elaborate on why the obtained results in the error tables give different optimum results for the mean error and differrent for the std.
Author Response
Thank you for commenting on our article, which provides valuable feedback on our proposed article. To simplify the review process, we have divided the response into three parts (one for each question) and also made changes to the manuscript to address these questions. These proposed changes appear highlighted in yellow in the annexed file.
Question a) Is the database used publicly available?
Comment: Yes, upon request to the authors.
Action: we have updated the paper to make this clearer in section 4.
Question b) How can Graph Convoluted Networks (GCN) be employed in this problem in the future?
Comment: Graph Convoluted Networks can potentially be employed on this problem to substitute the shortest path approach, creating a more sophisticated approach (please note that the focus on this paper was more on the representation learning inputted to the shortest path, not on the shortest path itself). However, like the deep learning approaches developed in the state of the art, they are a more complex form of Deep Learning and might require a larger
more comprehensive dataset for training.
Action: We added a remark to reflect the possibility of using GCN in future work in section 6.
Question c) Please elaborate on why the obtained results in the error tables give different optimum results for the mean error and different for the std.
Comment: Please note that average and standard deviation capture different quality aspects of a predictor. For instance, it is trivial to have a predictor with low (even zero) variance but that is always terribly wrong. The typical challenge is to simultaneously get a low bias and a low variance. Bias and variance capture complementary properties of the predictor, and all combinations of high / low values are possible in practice.

Reviewer 2 Report
In this paper, the authors replaced the Sobel filter with a novel neural network solution to improve breast contour detection based on the shortest path. This method is robust enough to deal with different scenarios that differ from its training standpoint. Overall, this paper is detailed, the experimental explanation is clear, and the effectiveness and generalization of this method were proved by experiments.
However, the following issues still exist:
1. In line 124 of page 3, " Silva et al. also proposed a hybrid approach to keypoint detection", it is suggested the authors to insert the corresponding references here.
2. This article is not innovative enough. It just makes a few changes to the RINDNET model and integrates the Soble operation into U-Net.
3. The experimental is not sufficient. The improved method in this paper should be compared with the existing state-of-the-art models.
Author Response
Thank you for commenting on our article, which provides valuable feedback on our proposed article. To simplify the review process, we have divided the response into three parts (one for each question) and also made changes to the manuscript to address these questions. These changes appear highlighted in yellow in the annexed file.
Question 1. In line 124 of page 3, " Silva et al. also proposed a hybrid approach to keypoint detection", it is suggested the authors insert the corresponding references here.
Action: The reference was added as rightly requested.
Question 2. This article is not innovative enough. It just makes a few changes to the RINDNET model and integrates the Sobel operation into U-Net.
Comment: This article provides innovation in the specific application it is designed for (it performs better than the previous state-of-the-art methods developed by Cardoso et Al., Silva et Al. and Gonçalves et Al.. Furthermore, it uses a new hybrid approach that couples the advantages of a conventional model with the capabilities of Deep Learning models that can learn effectively from data to improve results. This approach is useful as the models for this application have a small dataset for training. This work provides a new Deep Learning model of keypoint detection that can be quickly applied to different datasets, unlike the previous models.
The models proposed benefit from the same intuitive knowledge that informed the conventional method design while remaining a Deep Learning model that can learn from data to improve its performance. As more training data is collected we can easily retrain these models and it is expected they can learn more effectively and further improve their performance.
Action: Section 6 was enriched to reflect this more clearly.
Question 3. The experimental is not sufficient. The improved method in this paper should be compared with the existing state-of-the-art models.
Comment: The experimental evaluation is done in the available datasets where our models surpass current state-of-the-art performance. The first experiment is performed within the datasets used in the previous models. The second experiment is a cross-dataset experiment to understand how these models will react under new conditions. Additionally, we integrated the state-of-the-art edge detection model RINDNet to evaluate whether current Deep Learning edge detection models can be used to improve performance. As far as the knowledge of the authors goes, there is no other competitive approach published in the literature for the target application.

Round 2
Reviewer 1 Report
The authors have successfully addressed the revisions requested.
Author Response
Once again, thank you for commenting on our article, as it provided valuable feedback to our proposal. We have made minor changes to the experimental part of this article in order to address concerns expressed by another reviewer. All changes appear highlighted in yellow.

Reviewer 2 Report
1. It's better to introduce the RINDNet and other previous works in section 2 rather than in section 3.
2. The comparative experiment is not sufficient, and the ablation experiment is needed.
Author Response
Once again, thank you for commenting on our article, as it provided valuable feedback to our proposal. To address your first concern we rearranged the article's sections and the RINDNet model is now introduced in section 2: Previous Works. Additionally, we conducted more experimental evaluations on different edge detectors. In doing so, we found no edge detector that can improve upon the Shortest Path results using the Sobel operator and Section 6 highlights that effort. This effort has contributed to a more in-depth experimental process by comparing our models against the most commonly used edge operators. It is worth noting that when performing these experiments we found new results for the 5-fold cross-validation for Sobel as we had used a different evaluation technique before and that inconsistency is now corrected. Despite that, these new results don't alter our overall conclusion as the Sobel operator continues to be outperformed by our proposed method.
Article changes appear highlited in Yellow.
